# Public–Private Partnerships: A Fresh Risk-Based Approach to Water Sector Projects

**Sónia Lima** [1], **Ana Brochado** [1,*] **and Rui Cunha Marques** [2]

1   Centro de Estudos sobre a Mudança Socioeconómica e o Território (DINÂMIA'CET), ISCTE—Instituto Universitário de Lisboa, 1649-026 Lisbon, Portugal
2   Civil Engineering Research and Innovation for Sustainability (CERIS), Instituto Superior Técnico, University of Lisbon, 1049-001 Lisbon, Portugal
*   Correspondence: ana.brochado@iscte-iul.pt

**Abstract:** This study focused on the unbalanced relationships that can arise in current public–private partnership (PPP) risk management frameworks, especially in developing countries' water sectors. Different stakeholders' perceptions of risk management were examined by analyzing 15 interviews in Portugal and Mozambique. The hybrid method included semantic, descriptive statistic, content, and narrative analyses. To achieve the research objectives, the semi-structured interview transcripts were processed using quantitative and qualitative techniques to collate relevant actors' opinions of risk management in PPP water projects. Five risk categories were identified. The interviewed experts ranked the financial risk category as the most crucial, followed by infrastructure, commercial, technical and operational, and context risks. However, when the transcripts were evaluated from a risk factor perspective, the context risk category unexpectedly jumped to first place. Twenty-five high-impact risk factors were isolated in the semi-structured interview contents. The top five most critical risk factors were political interference, no performance measurement baselines, an unfavorable private investment climate, nonpayment of bills, and water assets uncertain condition. The results comprise a fresh contribution to the existing knowledge about experts' perceptions of PPP contract risks, including that prior research and specialists categorize financial risks as the most important. The findings further reveal that experts consider managing context risks to be the key factor in PPPs success in developing countries, as well as highlighting the need to explore these risks more fully in emerging economies' water sectors. In addition, a complete risk management cycle is proposed based on the interviewed professionals' opinions, in which risk assessment and risk treatment or mitigation measures are dealt with simultaneously.

**Keywords:** public–private partnership (PPP); water sector; developing country; risk management

## 1. Introduction

In 2021, investment in public–private partnership (PPP) projects reached USD 76.2 billion, allocated to 240 ventures. The water sector usually registers low levels of investment, but this area of PPP contracts have also registered the largest commitment of funding in a decade: USD 9.9 billion (13% of all investment) [1].

Risk management in PPP projects is a compelling concern among experts in this field. The academic literature shows that this topic is currently quite hot. Cui et al. [2] also provide evidence that risk management is a key issue in PPP research given that risk management and success factors were ranked fourth in importance and that 18% of this study main domains included PPP contracts. In related research covering a sample of 37 publications, 13 paid much attention to PPP risk management approach in the water sector [3]. Most previous studies on this topic have focused on risk identification and only to a lesser degree on risk analysis, e.g., [4,5].

Risk management frameworks can be broken into two critical phases: risk assessment and risk treatment or mitigation measures. The first phase includes risk identification,

analysis, and evaluation, so the final outputs should provide a list of critical risk factors. The assessment stage thus clarifies how the identified risks can affect contracts' objectives and performance. Appraisals necessarily comprise risk analysis and evaluation, yet researchers have failed to develop complete risk assessment models that allow critical risk factors to be identified more fully and thus ensure the second phase of risk treatment or mitigation produces useful findings. The latter stage implements measures and management strategies that include quantifying impacts and minimizing effects [2,4,6,7].

No consensus has been reached on the best methodology or terminology to use in risk assessments. According to the existing international standards, risks can be classified into categories based on the type of threat [7]. Putting risks into groups can facilitate the discrimination between and identification of these factors. Unkovski and Pienaar [8], for instance, proposed that risks can be categorized as financial, legal, and technical. Projects in controlled environments (i.e., PRINCE2) and other standard methodologies have been used to define risk factors [9]. Risk identification is accomplished by conducting systematic observations to classify specific projects' potential risks.

Risk factors are easily affected by external forces such as social and cultural diversity or PPPs location in developed or developing regions. These factors can quickly change in response to socioeconomic contexts substantial influence. Risk evaluation must, thus, reflect the likelihood of particular events occurring multiplied by the corresponding quantified impacts [6].

Risk allocation is also crucial for researchers to understand which partner (i.e., private or public) is responsible for addressing the identified risks (i.e., the risk owner) [10,11]. In the water sector, the biggest risk factors are non-economically viable water rates, water price uncertainty, financing, tax policy changes, interest rates and their volatility, and water resource price instability. Other factors are unstable governments, breaches of contract, weak national financial institutions, and resistant public opinion [12].

The process of ranking risk factors produces another relevant category: critical risk factors [13]. Risk evaluation and ranking have both attracted scholars' attention in recent years [13,14]. The water sector characteristics mean that risk factor rankings cannot be generalized to all settings [12]. For example, contracts range and chronology affect the expected risks directly related to the design and construction phases (e.g., planning and technology issues, plants delays or low performance, and total cost overage), while other risks are linked to operational and maintenance risks (e.g., market demand, operations, and supply issues). Finally, one group of risks is common in all phases (e.g., economic externalities, natural disasters, environmental issues, and macroeconomic variables including inflation, interest rates, and currency fluctuation). Researchers most frequently mention external economic and institutional factors such as governments' breaches of contract, natural catastrophes, water resources, financing, water price uncertainty, water rates, price instability, and poor performance [10,15,16].

According to the literature, other risks also need to be considered, for instance, risks related to foreign exchange rates, corruption, water theft, nonpayment of bills, political interference, high operational costs, pipeline failures, a lack of PPP experience, and inflation rate volatility. Additional risks include construction time and cost overruns, poor contract design, political discontent, early contract termination, poor design and construction problems, intra-partner conflicts, land acquisition issues, and public opposition to PPP contracts [4].

The second phase of risk management frameworks is based on the critical risk factors identified in the assessment stage that determine the most appropriate risk treatment or mitigation measures, including risk management strategies. The existing literature mentions two main types of solutions: acceptance versus treatment or mitigation measures. Accepting risks occurs when public partners evaluate possible risk responses and rationalize doing nothing about the emerging risks based on this strategy's economic and social advantages. In contrast, risk treatment or mitigation measures offer a range of solutions [17–19].

Risk transfer (e.g., contracting insurance policies) is one valid instrument with which to achieve value for money [20,21]. In addition, awareness of PPPs level of risk means that reducing expected risks allows partners to take action, such as revising projects dimensions and scope to reduce risk exposure and attract private participation [22]. Risk allocation also needs to be reviewed to ensure that all identified risks have at least one owner and that each PPP contract risk is assigned to the partner best able to manage it (i.e., risk-sharing options) [23]. Risk management further entails selecting appropriate tools that increase partners' control over critical risk factors and future threats [24].

Scholars have based their results mainly on experts' opinions in order to identify risk factors and critical risks, e.g., [11,25]. For example, Marques [26] conducted a study of PPP arrangements in Brazil and concluded that the risk matrix was unbalanced because most risks were allocated to the public sector.

The extant literature provides evidence that, traditionally, scholars have stopped short of constructing solutions that meet the challenges of risk treatment or mitigation. For example, of the 13 aforementioned studies, 70% did not address risk treatment or mitigation issues. Proposed solutions have included improving the available monitoring, supervising, and reviewing tools, which have become part of the increasingly trendy research stream focused on improving risk management [27]. PPP experts have, however, called for additional studies on risk mitigation mechanisms [4].

The present research specifically sought to examine risk management in developing countries PPP water contracts more fully by applying a more holistic approach to the two main risk management phases—risk assessment and risk mitigation measures—and targeting a developing country in Africa: Mozambique. Two questions were addressed:

1.　What are the most important risk categories in PPP contracts, according to experts?
2.　How can critical risk factors be mitigated?

The remainder of this paper is structured as follows. The next section describes the methodology (i.e., research context, data collection, and data treatment). The third section presents the main results, and the final section offers the conclusions.

## 2. Materials and Methods

### 2.1. Research Design

Assessing risk management frameworks more holistically is a challenging task. To this end, a hybrid methodology was applied: semantic analysis using a word cloud program, descriptive statistics generated by SPSS software, and content and narrative analyses. The research design was based on semi-structured interviews and their results.

Closed and semi-open questions were developed to address the research questions. The semi-structured interview protocol adopted had already been successfully used in previous studies [28,29]. The procedures were planned, constructed, and completed to ensure the questions were answered fully (see Figure 1).

The first step was to identify risk categories in the existing literature, which provided the basis for the interview guide. A systematic literature review was conducted to find and examine studies of water sector PPPs, water projects, and the associated risks that were published in English and listed in Scopus. A search was carried out for selected keywords in the abstract, keywords, and title of these documents.

The risk selection procedure was based on the results of an analysis of 37 studies that fulfilled three criteria: (1) focus on PPP water sector projects and risk, (2) a Q1 or Q2 classification by the SCImago Journal Rank indicator and Web of Science database for 2018, and (3) publication during the 21-year period defined (i.e., 1999–2020). The publications were reviewed and cataloged based on the following features: title, keywords, abstract, authors, author affiliations, geographical context (i.e., country or region), type of research (i.e., empirical, theoretical, or case study), type of PPP contract, method (i.e., qualitative, quantitative, or mixed), and a focus on either risk assessment or risk treatment and mitigation, or on both. The findings were stored in an Excel file.

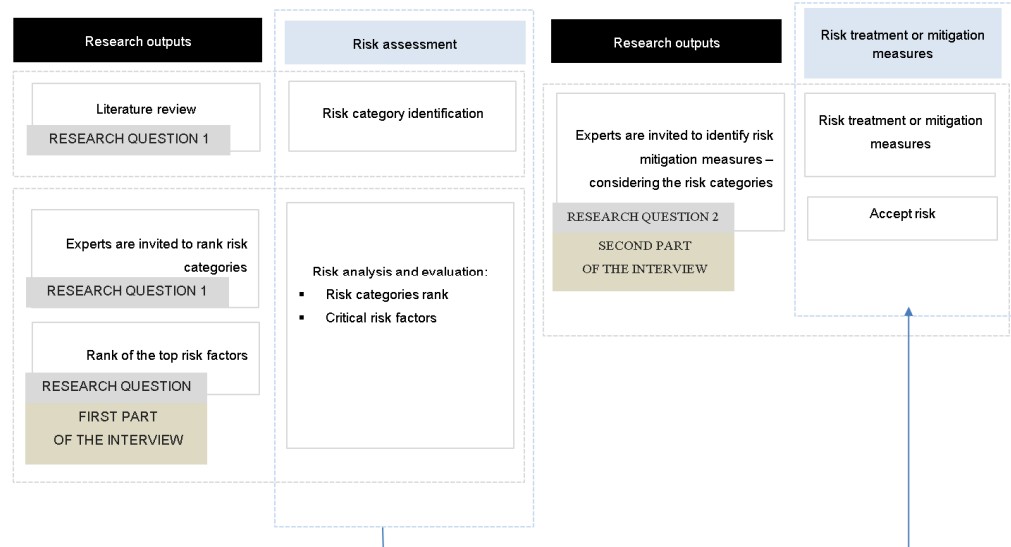

**Figure 1.** Risk management framework research process.

The semantic analysis was conducted using a word cloud generator program. To strengthen the results, the word cloud software extracted the data output's synonyms, antonyms, and similar words, which were then aggregated and entered into the Excel file's "Words" column. This procedure was performed via visual observation. The five identified risk categories include 30,559 words, or 19.24% of a total of 158,801 words (see Table 1).

**Table 1.** Risk category identification (37 studies).

| Risk Category | Words | Number of Words |
|---|---|---|
| Financial | Financial | 14,054 |
| | Investment | 854 |
| | Economic | 587 |
| | Cost | 441 |
| | Tariff | 336 |
| | Exchange rate | 153 |
| | Inflation | 134 |
| | Payment | 93 |
| | Tax | 87 |
| | Loan | 65 |
| **Subtotal** | | 16,804 |
| Context | Politics | 2382 |
| | Government | 1231 |
| | Regulation (sector) | 436 |
| | Legal | 244 |
| | Regional | 140 |
| | Corruption | 115 |
| | Socioeconomic | 122 |
| | Education | 15 |
| **Subtotal** | | 4685 |
| Technical and operational | Operational | 1909 |
| | Technical | 1647 |
| | Performance | 426 |
| | Structure | 232 |
| | Technology | 182 |
| **Subtotal** | | 4396 |

**Table 1.** *Cont.*

| Risk Category | Words | Number of Words |
|---|---|---|
| | Commercial | 1172 |
| | Contract | 862 |
| Commercial | Market | 377 |
| | Population | 196 |
| | Customer | 106 |
| **Subtotal** | | 2713 |
| | Infrastructure | 1083 |
| Infrastructure | Construction | 586 |
| | Design | 292 |
| **Subtotal** | | 1961 |
| **Total (subcategories)** | | 30,559 |
| **Total (word clouds)** | | 158,801 [a] |

Note: [a] Counted words taken from 37 studies focused on water sector risk management.

Financial risks were ranked as the most important, with 54.99% of the total words, followed by context (15.33%), technical and operational (14.39%), commercial (8.88%), and infrastructure (6.42%). Financial and economic risks have also always been a major problem, especially in water and wastewater projects [5,30]. These results facilitated the five risk categories integration into the semi-structured interview protocol.

The interviewees were subsequently invited to provide possible risk factors related to the assessment phase and suggest corresponding mitigation measures [6,31,32]. To avoid any ambiguity, definitions of the five risk categories were developed and read to the interviewees (see Table 2).

**Table 2.** Risk categories.

| Risk Category | Definition and Main Characteristics |
|---|---|
| Financial | Associated with the ability (or not) to secure the necessary funds from both partners for the success of the PPP projects. |
| Context | Related to the background, political and social–cultural, and economic background elements that can have an impact or constrain the PPP projects. |
| Technical and operational | Technical risks and operational issues that can affect (positively or negatively) the project success. It is connected to the PPP performance and its ability to provide the service in a timely and efficient way. |
| Commercial | Linked to the commercial provision of water supply services to customers, including the collection capacity in PPP projects. |
| Infrastructure | The impact that a good or bad preservation and awareness of PPP assets can have on the success of the project outcomes. |

The interview results facilitated the successful completion of the first stage of this risk management framework study. The experts interviewed were asked semi-open questions to identify the most important risk factor categories and then invited to define the most critical risks and potential treatment or mitigation measures. Their responses thus guided the choice of which categories were discussed in the second part of the semi-structured interviews.

The interviews started with an explanation of the project primary goals. The researchers then asked the interviewees to talk about their PPP contracts and/or industry experience. The interview guide comprised four main sections. First, the participants ranked the five most critical risks identified in the systematic literature review according to their perceived importance. The review's findings ensured the interview results had content validity. Second, the interviewees identified the main risk events associated with the critical risk categories while keeping in mind PPP contracts design and operationalization.

Third, the participants suggested mitigation measures for each risk group. Last, the experts interviewed provided data on their professional experience.

The participants were then prompted to identify other potential interviewees (i.e., snowball sampling) [33,34]. To ensure the results reliability, the interviews were transcribed and independently coded by the three researchers, and the final version was based on a consensus.

### 2.2. Experts' Profiles

The experts were considered eligible for participation in this study if they already had extensive experience in working with PPP contracts in developing countries and with governments, sector regulators, and utility companies in Mozambique. The semi-structured interviews were conducted by the research team in Mozambique (14) and Portugal (1).

The researchers conducted 15 semi-structured interviews to gather data from a broad, heterogeneous sample. The sampling methods included judgmental (5 recruits) and snowball (10 recruits) procedures. Similar studies have successfully applied these approaches [12].

The interviewees comprised top management (5), technical experts (3), directors (2), coordinators (2), advisors (2), and a consultant (1). The experts held senior-level positions, and 80% had worked in the public sector, mainly in the industrial and water sectors, and 20% in the private sector. The interviewees' relevant work experience, background, and organizational affiliations ensured their opinions were reliable [4].

### 3. Results

PPP contracts are based on the principle of building a partnership between public and private organizations. Both are expected to fulfill these long-term contracts, which are a vehicle to develop, rebuild, or maintain complex infrastructure, thereby increasing services efficiency and thus clients' social well-being [32]. The private partner should be able to count on immediate compensation if significant changes are made to a PPP water contract. The current research interviewees ranked political interference as one of the top five most critical factors.

The public partners' main function is to control and monitor the private partners' activities. This responsibility starts even before the bidding phase, as the government agencies involved need to correctly define the relevant contract objectives, investments, and economic and financial implications, including the best PPP model to be applied. The public partners' obligations must also include infrastructure maintenance or a clear description of the contract stipulations regarding related issues.

The literature review revealed that 13 of the 37 studies were relevant in terms of PPP risk management processes in the water sector. The scholars based their results mainly on experts' opinions regarding risk factors and critical risks, but their research did not consistently include proposing and evaluating risk treatment or mitigation measures. More specifically, nine studies out of thirteen (70%) did not address risk management issues, although all the authors made recommendations and suggested possibilities for future research.

Notably, the water sector characteristics mean that its critical risk factors cannot be generalized to different contexts. This sector is especially complex as it can present considerable diversity in the project type and external environment, among other direct or indirect determining factors. The risk factors in the context of PPP project thus need to be identified separately for each project according to the sector and surrounding environment.

The content analysis found many possible risks and risk factors connected to PPPs in the water sector in the 37 studies included in the systematic literature review. Visual observation was used to isolate 365 related text segments. After removing duplicates and similar wording and meanings, 122 risk factors were identified and integrated into the semi-structured interviews to help the participants recall potential risk factors found in Mozambique's water sector. The interview results cover 25 risk factors (see Table 3).

**Table 3.** Risk factor list (15 interviewees).

| Risk Factors | Risk Factor Frequency |
|---|---|
| Absence of policy and legal frameworks | 2 |
| Government officials' abuse of power | 7 |
| Inter-partner conflicts | 4 |
| Construction time and cost overruns | 2 |
| Continuous monitoring | 5 |
| Corruption | 4 |
| Design and construction deficiencies | 4 |
| Employee theft | 5 |
| Insufficient project financing supervision | 7 |
| Macroeconomic inconsistencies | 6 |
| No performance measurement baselines | 19 |
| Nonpayment of bills | 14 |
| Operating and maintenance cost escalation | 9 |
| Inadequate planning | 6 |
| Political interference | 26 |
| Poor contract design | 8 |
| Procurement risks | 3 |
| Regulatory risks (weak regulations) | 11 |
| Climate change | 4 |
| Technical leakage issues during distribution | 1 |
| Overall unfavorable private investment climate | 18 |
| Water assets uncertain condition | 10 |
| Water pricing and tariff review uncertainty | 3 |
| Water theft | 5 |
| Public and private partners' weak capabilities | 1 |
| Total | 184 |

As mentioned previously, five risk categories were identified in the literature review: financial, context, technical and operational, commercial, and infrastructure risks. This list was integrated into the interview protocol. The experts suggested possible risk factors (i.e., the risk assessment phase) and risk treatment or mitigation measures for all five risk categories. The results included 25 factors (184 responses) and 38 measures (148 answers) (see Table 4).

**Table 4.** Results for risk assessment and risk treatment or mitigation measures.

| | Number of Factors | Frequency (Answers) |
|---|---|---|
| Risk factors | 25 | 184 |
| Risk treatment or mitigation factors | 38 [a] | 148 |

Note: [a] 38 different types of risk treatment or mitigation measures identified that can be associated with risk factors.

### 3.1. Risk Categories Ranking

3.1.1. Closed Questions Results

The first part of the semi-structured interviews was based on closed questions (i.e., the structured interview). The interviewees were invited to rank the five risk categories by importance. The descriptive statistical analysis resulted in an overall ranking of the categories (see Table 5). The scores were based on a Likert scale ranging from 1 ("Less important") to 5 ("Very important").

The aggregated results included that 27% of the answers in the "Very important-Important" range (i.e., 5 and 4 on the Likert scale) ranked the infrastructure risk category first. Financial and commercial risks came second (23% and 20%, respectively), followed by context (17%) and technical and operational (13%).

The responses with "Moderately important" scores (i.e., 3 on the Likert scale) placed commercial risks first (40%), followed by financial (27%), technical and operational (20%), and infrastructure (13%). The context risk category failed to appear in these answers.

**Table 5.** Risk category ranking descriptive statistics results.

| Risk Category | Mean | Mean Rank | Median | Mode | Standard Deviation |
| --- | --- | --- | --- | --- | --- |
| Financial | 3.27 | 1 | 3 | 3 and 5 | 1.49 |
| Commercial | 3.27 | 1 | 3 | 3 | 1.10 |
| Infrastructure | 3.27 | 1 | 4 | 4 | 1.22 |
| Technical and operational | 2.79 | 4 | 3 | 2 | 1.31 |
| Context | 2.53 | 5 | 2 | 1 | 1.85 |

The final set of combined values, namely, "Slightly important-Less important" (i.e., 2 and 1 on the Likert scale), showed that 35% of these interviewees consider context risks to be the least important category. The remaining responses mentioned technical and operational risks (24%), infrastructure (17%), financial (14%), and commercial (10%).

The above results did not sufficiently clarify the interviewees' opinions, so additional quantitative analysis (i.e., descriptive statistics) was conducted using SPSS software (see Table 5 above). The mean values placed three risk categories in first place: financial, commercial, and infrastructure risks. The financial risks overall mean was 3.27 (standard deviation = 1.49; means between 2 and 5), which put this category at the top, followed by infrastructure (3.27 $\pm$ 1.22; means from 2 to 4) and commercial risks (3.27 $\pm$ 1.10; means between 2 and 4). Financial risks top ranking confirmed previous studies findings.

Infrastructure risks came second. When the earlier results were combined with the median (4) and mode (4) outputs, the infrastructure risk category overall was scored higher than the commercial risks given the latter lower median (3) and mode (3) values. The technical and operational risk category was ranked fourth with a mean of 2.79 $\pm$ 1.31 (i.e., means between 1 and 4), a median of 3, and a mode of 2. Finally, context risks were assigned scores with a mean of 2.53 $\pm$ 1.85 (i.e., means from 1 to 4), a median of 2, and a mode of 1.

### 3.1.2. Risk Category Ranking from a Risk Factor Perspective

The interviewees together identified 25 risk factors. Each expert was invited to suggest risk factors for each risk category and express his or her opinion about these groups' significance. The categories that were ranked the highest were context risks with 53 mentions (29%) and financial with 47 (25%). Commercial risks came third with 35 references (19%), and the fourth- and fifth-ranked categories had similar results: infrastructure with 25 (14%) and technical and operational with 24 (13%) (see Figure 2).

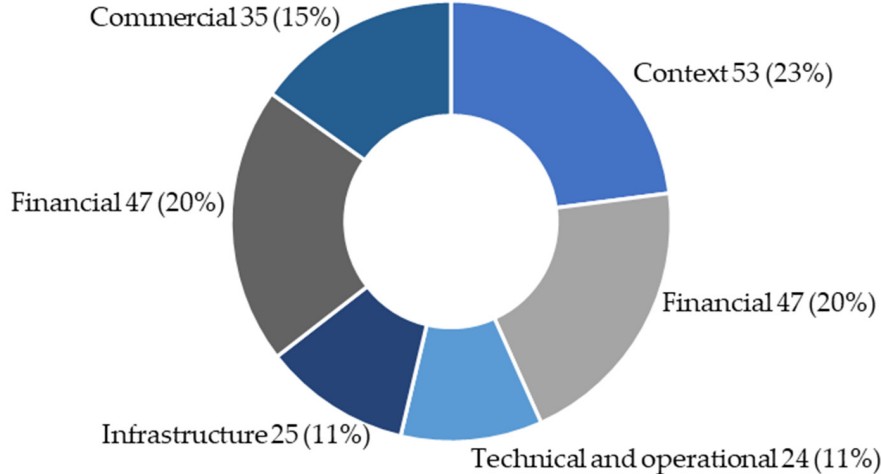

**Figure 2.** Risk factors and risk categories.

The above results were compared with the previous risk category rankings and closed-question responses. Significant differences were detected, except for financial risks, which only dropped from first to second. In contrast, the context risk category once more presented the most contrasting position in the ranking, as context risks had originally been placed last. A comparative analysis of the experts' assessments of the latter factors confirmed that this category was mentioned the most often, at 53 mentions (29%).

*3.2. Critical Risk Factors Identification*

The experts' initial 184 mentions and 25 identified risk factors were next ranked by their frequency. The results highlighted the top five risk factors, which are hereafter referred to as critical risk factors (see Table 6).

**Table 6.** Critical risk factors.

| Critical Risk Factors | Top Risk Category | Risk Factor Top Frequency |
|---|---|---|
| Political interference | Context | 26 |
| No baselines for performance measurement | Commercial | 19 |
| Unfavorable global private investment climate | Financial | 18 |
| Nonpayment of bills | Commercial | 14 |
| Water asset condition uncertainty | Infrastructure | 10 |
| Total | | 87 [a] |

Note: [a] Final total of 87 mentions out of the initial 184 separate references.

The above findings show that the "political interference" risk factor is the top risk factor with 26 mentions, in which this factor is mostly put into the context risk category (i.e., 19 times) (see Table 6 above). The second most critical risk factor is "no performance measurement baselines", with nineteen references, which mostly classified this factor as a commercial (nine) or financial (seven) risk. The third factor is an "overall unfavorable private investment climate," with eighteen mentions, which placed it in the financial (thirteen) or context (four) risk categories. In fourth came "nonpayment of bills", with fourteen allusions, which mostly classified this critical factor as infrastructure (nine) risk.

*3.3. Critical Risk Factors and Treatment or Mitigation Measures*

Five critical risk factors were thus identified in the risk assessment phase. The following subsections discuss these factors in more detail, as well as the interviewees' opinions regarding possible risk treatment or mitigation measures.

3.3.1. Political Interference

The political interference risk factor was considered the most important in the experts' responses (i.e., 26 mentions) during the interviews risk assessment phase. The participants identified this factor as a component of all five risk categories. According to the interviewees, this critical risk factor can have a transversal impact on PPP projects results. Political interference was mostly put into the context risk category (nineteen references), followed by financial (three), commercial (two), and infrastructure and technical and organizational (both with one). The existing literature also considers political interference to be a critical risk factor in PPP contracts [25,35].

The present interviewees subsequently identified measures to reduce this critical risk factor during the risk treatment or mitigation phase. Developing countries water costs are strongly affected by political and macroeconomic instability, which confirms that the context risk category is the most important. In particular, PPP water contracts need governments to make a political commitment to ensure that water rates reflect operational costs in order to maintain private parties financial sustainability, as the latter have little or no control over political interference [35]. Governments are responsible for providing a functional framework that ensures the right tools are introduced to encourage gradual

utility rate increases [36]. If substantial changes in PPP projects are necessary, the resulting costs should be fairly transferred to users or directly covered by the government.

Political interference can thus restrict regulators and private operators activities (e.g., rate adjustments) [35]. One of the experts interviewed for the present study argued that "a robust legal framework [needs to be created] to ensure the independence of the sector regulator, increasing its power to intervene in the water sector and PPP contracts" (personal communication). Unjustified political interference and weak government commitment reduce the water sector attractivity to the private sector, especially in the case of PPP contracts that, by definition, should be medium- or long-term [37].

### 3.3.2. No Performance Measurement Baseline

This critical risk factor was ranked second in the experts' answers, with 19 mentions. Similarly to political interference, this factor is a transversal issue, fitting into most risk categories [38]. In the current research, no performance measurement baseline appeared in all the risk groups except for infrastructure risk. The interviewees mostly identified this factor as a commercial (nine references) and financial (seven) risk.

Ameyaw and Chan [10] also found that this critical risk factor is a determinant of PPP water contracts success in Ghana. The lack of baselines hampers effective assessments of the private sector performance, which can have a negative impact on intra-partnership relationships.

The present analyses identified two major potential sources of baselines: management and control mechanisms. One expert suggested that a possible mitigation measure to ensure good project management is "the creation of tools that ensure effective management skills" (personal communication). According to another interviewee, internal control can be maintained with "the introduction of a list of requirements regarding accounting management methods and registration, namely, the type of software that will store the documentation data" (personal communication). A third participant called for the definition of "adequate levels of service [quality, quantity, and accessibility] with which the private partner has to comply" (personal communication). Management skills and competencies can be compared to a list of minimum work experience and qualifications (e.g., a background including jobs in developing countries), which could be added as a mandatory requirement to be fulfilled by private partners of PPP projects.

Public partners need control mechanisms to monitor private partners' performance. These tools should be properly designed and addressed before the bidding phase begins [39]. Monitoring mechanisms are thus an important way to ensure PPP objectives are met.

### 3.3.3. Overall Unfavorable Private Investment Climate

This critical risk factor was ranked third in the experts' answers, with 18 mentions in the risk assessment phase responses. The interviewees identified this unfavorable climate factor as part of the financial (thirteen allusions), context (four), and infrastructure (one) risk categories.

Ameyaw and Chan's [10] results again confirm the present experts' ranking in the context of PPP risks in Ghana's water supply projects. The cited authors argue that an overall unfavorable private investment climate can reduce the chances of attracting good bidders. The current study interviewees similarly underlined the need to capture and retain higher bidders, specifying that risk mitigation measures need to "improve investment plans in order to reinforce the government ['s role] as the major endorser" (personal communication). Public partners are thus crucial to ensuring a good international reputation and increasing potential private partners' interest in PPP projects. This need is related to the financial risk category main objective, namely, raising the necessary funds to ensure PPP projects success.

A possible mitigation measure in this context is "the creation of an insurance policy to mitigate the risk of political interference (e.g., the Multilateral Investment Guarantee

Agency [MIGA])" (personal communication). Political risk insurance is a tool that has already been put into practice. Multilateral organizations such as the MIGA have already developed possible solutions, indicating that this kind of insurance is a valid way to mitigate and manage risks arising from adverse situations due to governments' intervention in developing countries.

### 3.3.4. Nonpayment of Bills

This critical risk factor was ranked fourth in the experts' responses, with 14 mentions in the interviews' risk assessment phase. Unpaid bills were allocated to the commercial (thirteen references) and financial (one) risk categories. Multiple scholars have previously confirmed that the nonpayment of bills is an important risk factor [4,22,40]. In addition, Marin [16] reinforces this assessment by alerting PPP project managers to possible unpaid bill risks. Legal constraints may not be in place to enforce water service payments, especially in developing countries, which can reduce private partners' expected revenues.

One of the current study interviewees asserted that "the introduction or reinforcement of the paying user principle" ensures that new contracts include direct and indirect mechanisms for billing customers (personal communication). Another expert said that private partners have to "introduce a budget to create awareness campaigns targeting their direct customers with the message: "It is necessary to pay to get access to basic goods as a way to get better levels of service" (personal communication). This strategy is thus a possible mitigation measure for the nonpayment of bills risk factor.

The adverse socioeconomic conditions of the population can jeopardize the expected profits, especially in developing countries, and compromise private partners' ability to provide adequate services. Additional measures may need to include, for example, pro-poor measures (e.g., lower average rates for users falling below the poverty line) and updated customer databases. These strategies can mitigate this critical risk factor and emerge as complementary solutions. In developing countries, another viable measure is the installation of pre-paid water meters [41]. The need to avoid unpaid bills is related to the commercial risk category main objective, that is, to ensure water supply services are provided to customers by including bill collection powers in PPP projects.

### 3.3.5. Water Assets Uncertain Condition

This critical risk factor was ranked fifth by the participants of the present research, with 10 mentions in the interviews risk assessment phase. The responses identified this factor as part of the infrastructure (nine references) and technical and operational (one) risk categories. Previous studies have also confirmed that water assets uncertain condition is important to PPP projects [4,22]. Service targets can be missed due to obsolete technology, equipment defects, poor maintenance, and inadequate asset repairs [10].

Regarding risk reduction measures, an interviewee mentioned "the introduction of mandatory clauses in the contracts, passing the responsibilities of infrastructure maintenance to the private partner" (personal communication). Another participant suggested "the creation of mechanisms that facilitate an adequate inventory of assets during the contract preparation phase" (personal communication). A third expert recommended "mechanisms that allow an external evaluation of infrastructure records accuracy" as mitigation measures (personal communication).

The uncertain water asset condition factor is thus closely connected to this research definition of the infrastructure risk category, which considers the impacts of good or bad states of preservation and an awareness of PPP assets effect on projects successful outcomes. Water infrastructure is complex to plan, construct, and maintain. Water services are thus characterized as having high sunk costs, and inadequate infrastructure management can have a significant impact on these projects' success.

## 4. Conclusions

To answer the first research question (i.e., what are the most important risk categories in PPP contracts according to experts?), a total of 37 studies with 158,801 words were examined, and five risk categories (30,559 words or 19.24%) were defined and integrated into the semi-structured interview guide. Fifteen experts were asked to rank the categories and, in a second stage, to provide more detailed information about the risks.

The interviewees also initially ranked the financial risk category as the most significant, followed by infrastructure, commercial, technical and operational, and context risks. Prior studies in various research contexts have, however, ranked these critical risk factors differently [2], although financial risk has been listed at the top in terms of probability of occurrence and detection [40] for water supply projects in developing countries such as Iran.

The present study five categories were subsequently assessed from a risk factor perspective. The most significant change in the ranking was in the context risk category, which went from last to first place. The results thus indicate that context risks merit additional attention and that this category could be a rich vein for future research to explore. This finding is in line with Ke et al.'s study [30], which identified government intervention as the most important risk factor as well as placing five government-related risks in the top ten. This type of macroeconomic risk [5] is country-specific [4], and it has a strong negative impact on PPP contracts because government interference decreases the private sector and investors' confidence [11].

The second research question (i.e., how can critical risk factors be mitigated?) was addressed after the interviews risk assessment phase, in which the experts identified 25 risk factors. The factors were ranked by frequency in the responses, and the top five were examined more closely. The political interference critical risk factor was considered the most significant, with 70% of the mentions placing this factor in the context risk category. The second most important was the no performance measurement baseline factor, which 47% of the interviewees' references put in the commercial category. The third was an overall unfavorable private investment climate, with 72% of the allusions assigning this critical risk factor to the financial category. The fourth was the nonpayment of bills factor, which was placed by 92% of the mentions in the commercial category. The last critical risk factor was water assets uncertain condition, and 90% of the experts' references included this risk in the infrastructure category. The results further show that the technical and operational risk category was not important in terms of the five critical risk factors.

The above results have clear theoretical implications. This study answered previous studies calls for more research on PPP risk management and success factors [2]. The present findings contribute to the existing knowledge about experts' perceptions of PPP contract risks. The surveyed literature and interviewed specialists ranked financial risks as the most significant risk category. The results also reveal that experts consider context risks associated with political and macroeconomic instability to be a key factor in PPP success in developing countries. Thus, the development of a robust legal framework is of utmost importance to ensuring water sector regulators' independence, minimizing unjustified political interference in PPP contracts, promoting transparency, and avoiding legal uncertainty and ambiguity.

This research also identified water sector risk mitigation measures, which is an under-researched area. Scholars have mainly based their results on experts' opinions about critical risk factors, but prior investigations have failed to provide the corresponding mitigation procedures. This lack of potential solutions in the literature highlights the value of the risk treatment or mitigation measures suggested by the present study participating experts for the top five risks. Political or government interference can be moderated by creating robust legal frameworks and restriction mechanisms. Control and monitoring tools should be used to maintain private partners' adequate performance levels to avoid future problems in PPPs. These mechanisms can be designed and incorporated into projects before the bidding phase as a practical solution for the no performance measurement baseline factor.

In addition, the risk of an overall unfavorable private investment climate needs to be minimized by, for example, confirming that governments will endorse the necessary investments. Political risk insurance can contribute significantly to transferring risk, which provides private partners with more security regarding major disasters and, especially in developing countries, adverse government actions, war, and terrorism. The fourth and fifth critical risk factors can be mitigated by pro-poor measures, updated customer and asset databases, and alternative collection methods, such as pre-paid water meters. These solutions thus help address the nonpayment of bills and water assets uncertain condition factors.

This study results also have managerial and societal implications. The present findings suggest that unwarranted government interference should be offset by appropriate measures. Water sector regulators' primary mission is to find the best national and international practices and incorporate them into contracts, which will lead to sustainable, consistent service quality improvement. Risk management clauses should additionally provide methods to reduce private partners' exposure to nonpayment of bills. These partners can alternatively be given the option to collaborate with the government and other stakeholders to develop awareness campaigns that raise low-income clients' awareness of the benefits of realistically priced, efficient water resources.

The above results are based on the interviewees' opinions, which can be seen as a limitation, even though the findings are supported by the existing literature, because a different sample could produce contrasting research outputs. The data were defined by a consensus between three researchers, which comprised a second shortcoming as the content analysis of the interviewees' responses was a subjective process. Regardless of these limitations, the analyses of the semi-structured interviews produced results that adequately answered the predefined research questions. More studies are needed to clarify the proposed risk mitigation measures impact on PPP contracts in terms of bankability, with reference to the countries and PPP programs involved.

**Author Contributions:** Conceptualization, A.B., R.C.M. and S.L.; methodology, A.B., R.C.M. and S.L.; formal analysis, S.L.; investigation, S.L.; writing—original draft preparation, S.L.; writing—review and editing, A.B. and R.C.M.; supervision, A.B. and R.C.M. All authors have read and agreed to the published version of the manuscript.

**Funding:** This research received no external funding.

**Data Availability Statement:** The data presented in this study are available on request from the corresponding author.

**Conflicts of Interest:** The authors declare no conflict of interest.

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
