# Peer review of "Public–Private Partnerships: A Fresh Risk-Based Approach to Water Sector Projects"

_infrastructures, doi:10.3390/infrastructures8060102_

Round 1

Reviewer 1 Report

do not repeat PPP in the keywords

check English with grammarly.com or similar software

introduction too long

refer also to project financing

ùwhy did you choose Portugal and Mozambique (former colony)? Motivate

Extend the conclusions to other countries

what are the practical implications of Your study?

why is it interesting

challenge your research question in the discussion

improve the references

https://scholar.google.com/scholar?hl=it&as_sdt=0%2C5&q=Public-private+partnerships%E2%80%94a+risk-based+approach+to+water+sector+projects&btnG=

put some statistics about PPP

which is the impact of risk on bankability?

explain in deeper detail the interview process to support your methodology

Author Response

Reviewer #2:

do not repeat PPP in the keywords

We are grateful for all your constructive feedback, which helped us to improve our manuscript.

The authors are grateful for this comment.

The keywords are: PPP, water sector, developing countries, risk management.

The author´s remove the second full written expression PPP (see page 1)

check English with grammarly.com or similar software

The paper was proofreaded by a native professional. Please see the proofreading certificate. 

introduction too long

The authors increased the conclusion section, for a more balanced manuscript.

refer also to project financing

The authors add a line regarding the funding.

Why did you choose Portugal and Mozambique (former colony)? Motivate

The authors clarified the data collection/ fieldwork procedure:  “The experts were considered eligible for participation in this study if they already had extensive experience in working with PPP contracts in developing countries and with governments, sector regulators, and utility companies in Mozambique. The semi-structured interviews were conducted by the research team in Mozambique (14) and Portugal (1)”.

Extend the conclusions to other countries

The authors added the following sentence to the Introduction to highlight that the results ae country scpecific

“The present research specifically sought to examine risk management in developing countries’ PPP water contracts more fully by applying a more holistic approach to the two main risk management phases—risk assessment and risk mitigation measures—and targeting a developing country in Africa: Mozambique”.

what are the practical implications of Your study?

The authors developed the managerial/societal implications of the study in the conclusions (please the highlighted text).

why is it interesting

The authors are grateful for this comment. It was added into the abstract the following sentence: “The study provides an original contribution to the current knowledge and perceptions of experts regarding risk in PPP contracts. Literature and experts rank financial risks as the most relevant risk category. However, the reality is that expert´s truly consider other source: context risks are the key actor to a PPP success.”

challenge your research question in the discussion

The authors answred the two research questions I the Coclusion

improve the references

References added to the document:

Ramos, H. M., Morillo, J. G., Rodríguez Diaz, J. A., Carravetta, A., & McNabola, A. (2021). Sustainable water-energy nexus towards developing countries’ water sector efficiency. Energies, 14(12). https://doi.org/10.3390/en14123525

Ke, Y., Wang, S., Chan, A., & Cheung, E. (2011). Understanding the risks in China’s PPP projects: Ranking of their probability and consequence. Engineering, Construction and Architectural Management, 18(5), 481–496. https://doi.org/10.1108/09699981111165176

World Bank. (2021). Private Participation in Infrastructure - 2021 Annual Report. 90. https://www.annualreports.com/HostedData/AnnualReports/PDF/NYSE_ABT_2020.pdf

put some statistics about PPP

The authors agree with the reviewer comment. It was add into the introduction the sentence “In 2021, investment in PPP projects reached a total of US$ 76.2 billion, allocated to 240 operations. The water sector, which usually registers low levels of investment, had its highest commitment in a decade, with about US$ 9.9 billion (13%) (World Bank, 2021).” (see page 1).

which is the impact of risk on bankability?

Although the authors  have not addressed the impact of risk mitigation on bankability, we recognized that it merits future research (please see the last sentence of the methodology)

explain in deeper detail the interview process to support your methodology

The authors detailed in the section ‘research design’ the design of the guide of the interview and the sampling procedure.

Reviewer 2 Report

Thank you for your interesting research. The authors have done a lot of work, using literature review and expert interviews. However, the manuscript in its present form is not ready for publication and I suggest Major revision. Here are my major concerns:

1. Please use a more systematic literature review to explore candidate risk factors. Besides, the authors need more graphs, discussions and tables as current work on Table 1 and Table 2 is quite insufficient and weak, which makes me wonder whether the literature is genuinely carried out or not. 

2. Please use demographic statistics, pictures or semi-structure questionnaire to increase the length of expert interviews. Current work is insufficient and weak, too. It makes me wonder whether the interviews are genuinely carried out or not. 

3. Current risk factors are quite simple. Maybe you need a hierarchical structure: level-one factors, level-two factors and etc. 

Author Response

Reviewer #3:

Thank you for your interesting research. The authors have done a lot of work, using literature review and expert interviews. However, the manuscript in its present form is not ready for publication and I suggest Major revision. Here are my major concerns:

We are grateful for all your constructive feedback, which helped us to improve our manuscript.

The authors addressed all issues raised by reviewer #3, that allowed to improve the manuscript.

1. Please use a more systematic literature review to explore candidate risk factors. Besides, the authors need more graphs, discussions and tables as current work on Table 1 and Table 2 is quite insufficient and weak, which makes me wonder whether the literature is genuinely carried out or not. 

The authors are grateful for this comment. The following text was added to the document:

The first step was to identify risk categories in the existing literature, which pro-vided the basis for the interview guide. A systematic literature review was conducted to find and examine studies of water sector PPPs, water projects, and the associated risks, which were published in English and listed in Scopus. A search was carried out for selected keywords in these documents’ abstract, keywords, and titles.

The risk selection procedure was based on the results of an analysis of 37 studies that fulfilled three criteria: 1) focus on PPP water sector projects and risk, 2) a Q1 or Q2 classification by the SCImago Journal Rank indicator and Web of Science database for 2018, and 3) publication during the 21-year period defined (i.e., 1999–2020). The publications were reviewed and catalogued based on the following features: title, keywords, abstract, authors, author affiliation, geographical context (i.e., country or region), type of research (i.e. empirical, theoretical, or case study), type of PPP con-tract, method (i.e. qualitative, quantitative, and mixed), and a focus on either risk as-sessment or risk treatment and mitigation—or on both. The findings were stored in an Excel file.

The semantic analysis was conducted using a word cloud generator program. To strengthen the results, the word cloud software extracted the data output’s synonyms, antonyms, and similar words, which were then aggregated and entered into the Excel file’s “Words” column. This procedure was performed via visual observation. The five identified risk categories include 30,559 words or 19.24% of a total of 158,801 words (see Table 1).

2. Please use demographic statistics, pictures or semi-structure questionnaire to increase the length of expert interviews. Current work is insufficient and weak, too. It makes me wonder whether the interviews are genuinely carried out or not. 

The authors added more details about the guide of the interview (research design) and the profile of the participants (expert’s profile)

3. Current risk factors are quite simple. Maybe you need a hierarchical structure: level-one factors, level-two factors and etc. 

Please see Table 1

Round 2

Reviewer 2 Report

Thanks for your careful revision and persuasive response. The manuscript in its present form is of good quality and reaches the standard of Sustainability. Thus, I would recommend Accept if the authors could:

1. Increase the resolution ratio of Figure 1, Table 2, Table 4, Table 5, Figure 2, Table 6

2. Reduce the length of Section 4 Conclusion and remove some contents to Section 2 or Section 3. 

Author Response

Reviewer #3:

Thanks for your careful revision and persuasive response. The manuscript in its present form is of good quality and reaches the standard of Sustainability. Thus, I would recommend Accept if the authors could:

We are grateful for all your constructive feedback, which helped us to improve our manuscript.

1. Increase the resolution ratio of Figure 1, Table 2, Table 4, Table 5, Figure 2, Table 6

The authors improved the resolution of Figures and edited the Tables.

2. Reduce the length of Section 4 Conclusion and remove some contents to Section 2 or Section 3. 

In order to address this comment the authors removed the following sentences:

 “….was addressed by identifying five risk categories based on aggregated data collected with a systematic literature review and word cloud program. The analysis included the aggregation of synonyms, antonyms, and similar words. The categories were detected via visual observations.” – page 13

“….Financial risks were ranked as the most important, with more than half the counted words, followed by context, technical and operational, commercial, and infrastructure risks.” - page 13

Moreover, the authors the following sentences to section 3

PPP contracts are based on the principle of building a partnership between public and private organizations. Both are expected to fulfill these long-term contracts, which are a vehicle to develop, rebuild, or maintain complex infrastructure, thereby increasing services’ efficiency and thus clients’ social well-being [32]. The private partner should be able to count on immediate compensation if significant changes are made to a PPP water contract. The current research’s interviewees ranked political interference as one of the top five most critical factors.

The public partner’s main function is to control and monitor the private partner’s activities. This responsibility starts even before the bidding phase as the government agencies involved need to define correctly the relevant contract’s objectives, investments, and economic and financial implications, including the best PPP model to be applied. The public partner’s obligations must also include infrastructure maintenance or a clear description of the contract’s stipulations regarding related issues.  – page 6 and 7.
